# Thermoregulatory heat-escape/cold-seeking behavior in mice and the influence of TRPV1 channels

**Yuta Masuda[1,2], Riho Sakai[1,2], Issei Kato[1,2], Kei Nagashima[2]***

**1** Graduate School of Human Sciences, Waseda University, Tokorozawa, Saitama, Japan, **2** Body Temperature and Fluid Laboratory, Faculty of Human Sciences, Waseda University, Tokorozawa, Saitama, Japan

* k-nagashima@waseda.jp

**Data Availability Statement:** All relevant data are within the paper and its Supporting Information files.

**Funding:** •KN •the W-ARCHS Research Project (A) for Establishing a Research Hub for Human

## Abstract

The present study assessed heat-escape/cold-seeking behavior during thermoregulation in mice and the influence of TRPV1 channels. Mice received subcutaneous injection of capsaicin (50 mg/kg; CAP group) for desensitization of TRPV1 channels or vehicle (control [CON] group). In Experiment 1, heat-escape/cold-seeking behavior was assessed using a newly developed system comprising five temperature-controlled boards placed in a cross-shape. Each mouse completed three 90-min trials. In the trials, the four boards, including the center board, were set at either 36˚C, 38˚C, or 40˚C, while one corner board was set at 32˚C, which was rotated every 5 min. In Experiment 2, mice were exposed to an ambient temperature of 37˚C for 30 min. cFos expression in the preoptic area of the hypothalamus (POA) was assessed. In Experiment 1, the CON group stayed on the 32˚C board for the longest duration relative to that on other boards, and intra-abdominal temperature ($T_{abd}$) was maintained. In the CAP group, no preference for the 32˚C board was observed, and $T_{abd}$ increased. In Experiment 2, cFos expression in the POA decreased in the CAP group. Capsaicin-induced desensitization of TRPV1 channels suppressed heat-escape/cold-seeking behavior in mice during heat exposure, resulting in hyperthermia. In conclusion, our findings suggest that heat sensation from the body surface may be a key inducer of thermoregulatory behaviors in mice.

## Introduction

Thermoregulation in homeothermic animals consists of autonomic and behavioral processes [1, 2]. Autonomic processes in rodents in response to heat include vasodilation of the tail vessels and saliva secretion. Neurons in the preoptic area (POA) of the hypothalamus and anterior hypothalamus are critically involved in these responses and are activated by both local hypothalamic and peripheral body surface heating [3]. The afferent thermal pathway terminates in the median POA (MnPO) via the lateral parabrachial nucleus in the pons (LPB) and activates the medial preoptic area, including the medial POA (MPO) and ventromedial POA (VMPO) [4].

Sciences. •The funders had no role in study design, data collection and analysis, decision to publish, or preparation of the manuscript.

**Competing interests:** The authors have declared that no competing interests exist.

In contrast, in behavioral thermoregulation, heat-escape/cold-seeking behavior is a key thermoregulatory response to heat [5] and is rapidly initiated after heat exposure [2]. Satinoff and Rutstein reported that the autonomic cold response was impaired after the lesion of the anterior hypothalamus in rats but the behavioral response was retained [6], which denies an involvement of the anterior hypothalamus in the observed behavioral response. Additionally, the lesion of the LPB, receiving thermal inputs from the body surface, abolished heat-avoidance behavior in rats [7], indicating the necessity of only peripheral thermal signals to activate behavioral responses. A previous study reported that optogenetic stimulation of warm-sensitive neurons expressing brain-derived neurotrophic factor and pituitary adenylate cyclase-activating peptide in the POA induced cold-seeking behavior in mice [8]. Nevertheless, it remains unclear how thermal inputs from the core and/or periphery drive thermoregulatory behavioral responses to heat. Furthermore, the contribution of the behavioral response to thermoregulation has not been quantitatively estimated.

The isolated capsaicin receptor is referred to as the transient receptor potential cation channel subfamily V member 1 (TRPV1) channel [9]. TRPV1 channels are activated with a thermal threshold of $> 40 ^{\circ}C$, thereby inducing nociception. Caterina et al. [10] reported that TRPV1 knockout results in minor defects in physiological heat sensation. Vandewauw et al. [11] demonstrated that in addition to TRPV1, TRPM3 and TRPA1 are necessary for noxious heat sensation. However, Yarmolinsky et al. [12] reported that thermal stimulation of the oral cavity at temperatures of 36–43°C activated neurons in the trigeminal ganglion in mice. Notably, these responses were abolished in normal mice that were administered a TRPV1 antagonist as well as in TRPV1 knockout mice. These results suggest that TRPV1 channels play a role in warm sensation; however, the role of TRPV1 channels in thermoregulation remains unclear.

Subcutaneous injection of capsaicin results in desensitization of peripheral capsaicin-sensitive neurons (i.e., C and Aδ sensory afferent nerves) as a chronic response [11, 13]. Distinct to TRPV1 knockout and/or the administration of a TRPV1 antagonist, which only abolishes the function of TRPV1, capsaicin injection silences neurons expressing TRPV1 channels [14, 15]. Although the control rats prefer ambient temperature of 30°C, rats with capsaicin-induced desensitization of TRPV1 channels prefer higher ambient temperature (35°C) [13]. However, the desensitized rats exhibit avoidance behavior at 40°C, at which rectal temperatures surpass 41°C [16]. These results suggest that the desensitization of TRPV1 channels induces an upward shift of heat sensitivity to activate thermoregulatory behavior, resulting in hyperthermia. Indeed, rats and guinea pigs with the desensitization are unable to activate thermoregulatory heat dissipation mechanisms in ambient temperatures of 32–40°C [13, 17]. Systemic administration of large doses of capsaicin (>200 mg/kg) results in the impairment of hypothalamic function involved in thermoregulation [18–20] and non-specific damage to the peripheral nerves [14]. However, these results may indicate the involvement of neurons during thermoregulation, which could be desensitized by high-dose capsaicin.

In the present study, we compared heat-escape/cold-seeking behavior between normal mice and mice, in which TRPV1 channels were desensitized by subcutaneous capsaicin injection. We hypothesized that heat-escape/cold-seeking behavior would be activated linearly with heat intensity in normal mice, but this behavior would be blunted in desensitized mice, and the lack of this behavior would increase body temperature.

## Materials and methods

### Ethics statement

All procedures involving animals were conducted following the guidelines for the Care and Use of Laboratory Animals of the Ministry of Education, Culture, Sports, Science, and

Technology, Japan. The animal experiments were approved by the Institutional Animal Care and Use Committee of Waseda University, Japan (2021-A104). All experiments and methods were performed in compliance with relevant regulations and Animal Research: Reporting of *In Vivo* Experiments (ARRIVE) guidelines.

## Animals

Male C57/BL6 mice (n = 16; body weight, 28–35 g; age, 7–10 weeks) were used in this study. The mice were individually housed in plastic cages (16 × 26 × 13 cm) at an ambient temperature ($T_a$) of 28°C with a 12 h/12 h light-dark cycle (lights on from 07:00 to 19:00 h). Food and water were provided *ad libitum*.

## Surgery

Under 2% isoflurane inhalation anesthesia, mice underwent insertion of a temperature-measuring device (18.8 × 14.2 × 7.1 mm; nano tag®; Kissei Comtec Co., Ltd., Matsumoto, Japan) with a built-in integrated circuit temperature sensor chip in the abdominal cavity using sterile techniques to measure intra-abdominal temperature ($T_{abd}$). The mice were allowed to recover for a minimum of 10 days.

## Preparation of capsaicin and desensitization procedures

Capsaicin solution (10 mg/mL; Fujifilm Wako Pure Chemical, Osaka, Japan) was prepared and dissolved in a vehicle consisting of ethanol, Tween 80, and normal saline at a ratio of 10:10:80, as reported previously [21]. To minimize acute stress from capsaicin injections, mice were anesthetized by intraperitoneal administration of an anesthetic cocktail consisting of medetomidine hydrochloride (0.3 mg/kg), butorphanol (5 mg/kg), and midazolam (4 mg/kg) [22]. The capsaicin solution was injected subcutaneously at a dosage of 5 mL/kg (i.e., 50 mg/kg) in eight mice (CAP group) [21, 23, 24]. The vehicle alone (5 mL/kg) was injected subcutaneously in the remaining mice (control [CON] group). Anesthesia was maintained after the injection for at least 60 min. The second injection was repeated 2 days after the first injection in the same manner for each group. The injection procedure resulted in a 35% mortality rate within 7 days after the second injection. However, successful desensitization did not affect food intake, body weight gain, and spontaneous movement of mice after the injection.

## Assessment of desensitization by capsaicin

Desensitization of TRPV1 channels was verified using a previously reported method (i.e., eye wipe test) 7 days after the second capsaicin injection [25]. The method involved the application of 5 μL of capsaicin solution (152.7 mg/mL) or vehicle (5% DMSO, 10% Tween 80, 85% normal saline) to the conjunctiva of one eye of each mouse. Immediately after the application, the mice began to present with specific behaviors, including wiping their eyes with their paws. The behavior was recorded using a smartphone at 240 fps. The number of behavioral counts was quantified for a period of 60 s by a single experimenter who was blinded to the aim and protocol of the present study. The other liquid (i.e., liquid not applied 2 days previously) was applied to the contralateral eye 2 days later. Liquid (i.e., capsaicin solution or the vehicle) and eye laterality (i.e., right or left) on Day 1 were randomly selected. The difference in the number of eye-wiping behavior counts between the two tests was calculated.

## Assessment of heat-escape/cold-seeking behavior using a newly developed system: Experiment 1

We developed a new apparatus for assessing heat-escape/cold-seeking behavior in the present study (Fig 1). The system comprised of a Plexiglas box (height of box walls: 19 cm) surrounding five Peltier boards (10 × 10 cm) that were placed in the shape of a cross at the bottom of the box (Fig 1). The temperature of each board was individually controlled using units connected to a personal computer. Our previous system had the same temperature-controlled Peltier boards placed in line [5], which could not exclude the influence of mice preferring either end of the board. These effects have been observed in other studies [26, 27], and the present system was designed to minimize these effects.

Each mouse was familiarized with the system and was placed 2–3 times in the system whereby all areas were set at 32˚C. Experiment 1 consisted of four trials that were conducted during the light phase of the light-dark cycle (13:00–18:00 h) on 4 separate days. One trial was a control trial, in which all areas in the system were set at 32˚C. During the other three trials (Trials 1–3), the temperature of one Peltier board among Areas 2, 3, or 4 (Fig 1A) was set at 32˚C, and the temperatures of the other Peltier boards were set to 36˚C (Trial 1), 38˚C (Trial 2), or 40˚C (Trial 3). The two areas (Areas 1 and 5) were always set at either 36˚C, 38˚C, or 40˚C in Trials 1–3, respectively. We set the temperature of Area 1 at 36˚C, 38˚C, or 40˚C to evaluate the geological preference of mice. The order of the control trial and Trials 1–3 was randomized. The Peltier board selected to be set at 32˚C was rotated every 5 min for 90 min in Areas 2, 3, and 4. Locomotor behavior was recorded with a video camera placed 64 cm above the board. The areas where the mice rested and their locomotor behavior were analyzed using an image analysis software (Smart v.3.0; Panlab SL, Barcelona, Spain). We considered a mouse to be at rest when its speed was 0 m/s. The resting duration in the area set at 32˚C was

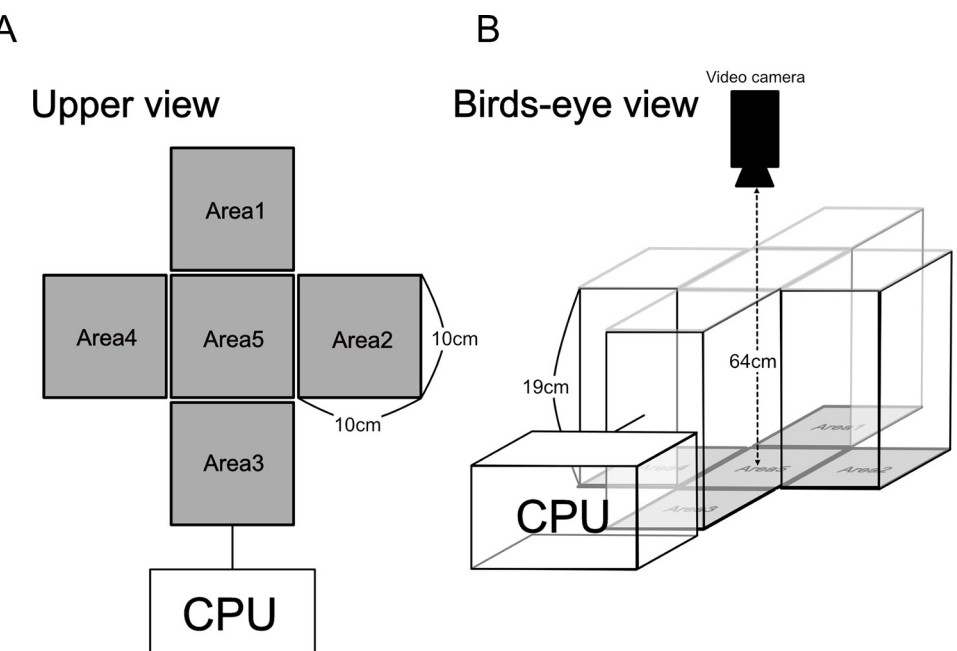

**Fig 1. Experimental system used for the assessment of heat-escape/cold-seeking behavior.** (A) Upper view. Five Peltier boards (10 × 10 cm) were placed at the bottom of the system in a cross-shape (Areas 1–5). The temperature of each board could be independently controlled using a computer. (B) Bird's-eye view of the system. The five areas were surrounded by Plexiglas walls with a height of 19 cm. The walls were painted black to minimize the influence of light entering horizontally.

summarized for each 30-min period. The percentage of resting time was calculated as the resting duration/30 min × 100. $T_{abd}$ was recorded every 12 s and averaged every 5 min.

## Assessment of core body temperature and c-Fos expression in the brain during passive heating: Experiment 2

After the completion of Experiment 1, mice were returned to their home cages and placed in a climate chamber set at $T_a$ of 28˚C for a minimum of 7 days. Then, mice were deprived of food and water and exposed to $T_a$ of 37˚C for 30 min starting at 10:00 h. $T_{abd}$ was monitored every 1 min. Three mice from each group were maintained at $T_a$ of 28˚C. Mice were sacrificed by cervical dislocation 30 min after the exposure and perfused via the left ventricle with normal saline followed by 4% paraformaldehyde (Sigma-Aldrich Japan K.K., Tokyo, Japan). The period of heat exposure was determined based on a previous report [28]. Mouse brains were immersed in 4% paraformaldehyde followed by 20% sucrose (Sigma-Aldrich Japan K.K.) in phosphate-buffered saline (PBS).

The brains were immersed in an optimal cutting temperature compound and frozen on crushed dry ice. Coronal sections of 25 μm thickness were prepared using a cryostat. After rinsing the sections five times with PBS, sections were incubated for 30 min in 0.3% hydrogen peroxide in PBS containing 0.3% Triton X-100. Then, sections were incubated with rabbit primary anti-c-Fos polyclonal IgG (1:4000 dilution; 9F6, Lot. 11; Cell Signaling Technology, Danvers, MA, USA) overnight. Subsequently, sections were incubated with biotinylated donkey anti-rabbit IgG (1:400 dilution; BA-100, Lot. ZG0818; Funakoshi Co. Ltd., Tokyo, Japan) for 120 min. The sections were incubated in avidin-biotin complex for 90 min after being rinsed in PBS and were, then, stained with 5% 3,3'-diaminobenzidine tetrahydrochloride in PBS. The sections were mounted on gelatin-coated glass slides and covered using covesrslips. After capturing digital images using a microscope, the number of cFos-immunoreactive (IR) cells was counted in three subregions of the POA: VMPO, bregma 0.40 mm; MnPO; and MPOA, bregma 0.14 mm (n = 5). Quantification was conducted for three consecutive sections and averaged using an automated method as previously reported [29]. The cFos-IR cell detection settings were as follows: binary threshold at a pixel intensity of 115, radius size of five pixels, and object circularity of 0.70.

## Statistical analysis

One-way analysis of variance (ANOVA) was performed to compare eye-wiping behavior counts and number of cFos-IR cells between the CAP and CON groups. A three-way repeated-measures ANOVA was performed to compare the percentage of resting time at 32˚C and $T_{abd}$ between the two groups. Post hoc tests were conducted using Bonferroni's method. A two-way repeated-measures ANOVA was conducted to compare $T_{abd}$ during heat exposure at 28˚C and 37˚C between the two groups. Post hoc tests were conducted using the Tukey (for one-way ANOVA) and Bonferroni (for other analyses) methods. Statistical analyses were conducted using Sigma-Plot statistical software (version 14.0; Systat Software, Milpitas, CA, USA). Data are presented as means ± standard deviations [SD]. The null hypothesis was rejected at $P < 0.05$.

## Results

### Verification of capsaicin desensitization

The desensitization of TRPV1 channels was evaluated using the eye-wiping test stimulated by local application of capsaicin. Successful desensitization was verified based on eye-wiping

behavior counts, which were lower in the CAP group (8 ± 5) than in the CON group (41 ± 10) (P < 0.001).

## Experiment 1

In the control trial, the percentages of resting time in Areas 1 to 4 (i.e., corner boards) were higher than those in Area 5 (i.e., the center board) (CON group: $F_{(4,20)}$ = 70.38, P < 0.001, 61 ± 12% and 2 ± 1%; CAP group: $F_{(4,20)}$ = 65.41, P < 0.001; 48 ± 16% and 1 ± 1%, respectively). Additionally, mice in both groups tended to stay on one board among Areas 1 to 4.

Fig 2 presents the percentage of resting times at 32˚C during the 36˚C, 38˚C, and 40˚C trials involving the CAP and CON groups. The percentage of resting time is presented for each 30 min period (A, B, and C) during the total trial period of 90 min. The percentage of resting time at 32˚C exceeded 33% in 30 min bins of each trial in the CON group. The percentages of resting times during the 38˚C and 40˚C trials were higher than those during the 36˚C trial in the CON group (83 ± 8%, 91 ± 5%, and 41 ± 1% at 31–60 min, respectively; P < 0.001). Additionally, the percentage of resting time at 32˚C observed at the period of 61–91 min became greater in the 40˚C trial than that in the 38˚C trial. During each trial, there were no differences in the percentages of resting time in Areas 2–4; however, the percentages of resting time in Areas 2–4 were greater than those in Areas 1 and 5 (39 ± 8%–18 ± 7% in Areas 2–4; 2 ± 2% in Area 1; and 0 ± 0% in Area 5 during the 38˚C trial; $F_{(4, 20)}$ = 113; P < 0.001).

In the CAP group, the percentage of resting time at 32˚C was lower than that in the CON group in each 30 min bin for each trial ($F_{(2, 44)}$ = 39.198; P < 0.001). Moreover, these values were similar among the three trials and measurement periods (16 ± 5%, 13 ± 7%, and 15 ± 8% at 0–30 min in the 36˚C, 38˚C, and 40˚C trials, respectively; P = 0.190). During each trial, there were no differences in the percentages of resting time among Areas 1–4 (14 ± 11%, 36 ± 22%, 25 ± 18%, and 12 ± 7% in Areas 1–4, respectively, during the 38˚C trial). The percentages of resting time in Areas 1–4 were greater than those in Area 5 (0 ± 0% in Area 5 during the 38˚C trial; $F_{(4, 20)}$ = 113; P < 0.001). During each trial, mice in each group rested on the board during the resting period. Behaviors such as lying on the board and body extensions were not observed.

Fig 3 presents $T_{abd}$ during the control (Fig 3A), 36˚C (Fig 3B), 38˚C (Fig 3C), and 40˚C (Fig 3D) trials in the CON and CAP groups. In the control trial, $T_{abd}$ remained unchanged in both the CON and CAP groups, with no between-group difference in values. In the CAP group, $T_{abd}$ values during the 36˚C, 38˚C, and 40˚C trials were higher than those during the control trial between 10 and 90 min; however, there were no significant differences in $T_{abd}$ in the CON group during trials (P = 0.885). In the CAP group, $T_{abd}$ was higher during the 40˚C trial than during the 36˚C and 38˚C trials (at 30 min, 40.5 ± 0.7˚C, 39.7 ± 0.3˚C, and 39.0 ± 0.3˚C for the 40˚C, 38˚C, and 36˚C trials, respectively; P < 0.001). $T_{abd}$ differed between the CON and CAP groups between 10 and 90 min during the 36˚C and 38˚C trials and between 5 and 90 min during the 40˚C trial (P < 0.001).

## Experiment 2

Mice in both groups were exposed to $T_a$ of 28˚C or 37˚C for 30 min; then, they were sacrificed. Fig 4 presents the $T_{abd}$ of the CAP and CON groups during exposure to $T_a$ of 28˚C and 37˚C. During exposure to $T_a$ of 37˚C, $T_{abd}$ of the CON group remained unchanged. $T_{abd}$ of the CAP group exceeded $T_{abd}$ of the CON group at 10 min and remained at a persistently higher level ($F_{(6, 56)}$ = 17.61; P < 0.001; 36.91 ± 0.86˚C and 38.66 ± 0.24˚C at 10 min in the CON and CAP groups, respectively). At $T_a$ of 28˚C, $T_{abd}$ remained unchanged in both groups, with no differences observed between the groups ($F_{(6, 42)}$ = 0.30; P = 0.93; 36.41 ± 0.9˚C and 36.49 ± 1.2˚C in

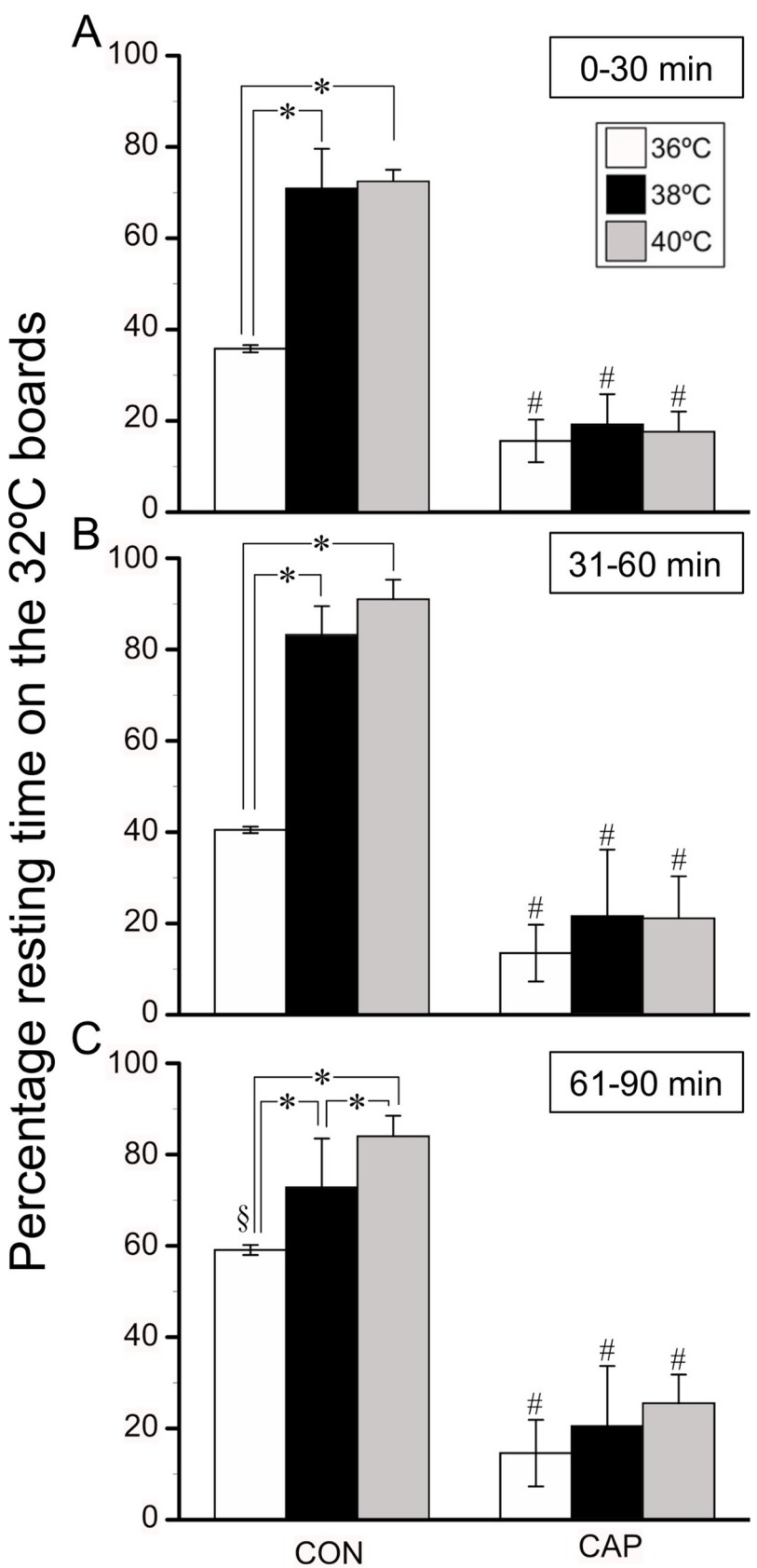

**Fig 2. Percentage of resting times at 32˚C during the 36˚C, 38˚C, and 40˚C trials in the CAP and CON groups.**
Percentage of resting time on the 32˚C board during the 36˚C (white bar; n = 8), 38˚C (black bar; n = 8 and 7, control
and capsaicin groups, respectively), and 40˚C (grey bar; n = 8 and 6, control and capsaicin groups, respectively) trials
among the control mice (CON group; left three bars) and capsaicin-desensitized mice (CAP group; right three bars) at
(A) 0–30 min (B), 31–60 min, and (C) 61–90 min. Values are presented as means ± standard deviations (SD).
*Significant difference between two corresponding trials, P < 0.05. #Significant difference between the CON and CAP
groups during the same trial and period, P < 0.05. §Significant difference between the values at 0–30 min and 31–60
min, P < 0.05.

the CON and CAP groups, respectively). $T_{abd}$ during housing condition ($T_a$ of 28˚C) at the
corresponding time of day was unchanged and similar to that in this experimental condition.

Fig 5 illustrates photoimages of two sections in the POA of the CON and CAP groups at $T_a$
of 28˚C and 37˚C. The counts of cFos-IR cells in the VMPO, MnPO, and MPO are summa-
rized in Fig 6. In the three areas, the counts were greater in the CON group at 37˚C than those
in the CON group at 28˚C and CAP group at 37˚C (VMPO; 11 ± 2 and 14 ± 4 at 28˚C and
49 ± 10 and 17 ± 3 at 37˚C in the CON and CAP groups, respectively, [$F_{(3,12)}$ = 36.77;
P < 0.001]; MnPO, 8 ± 6 and 9 ± 3 at 28˚C and 31 ± 2 and 14 ± 6 at 37˚C in the CON and
CAP groups, respectively, [$F_{(3,12)}$ = 22.27; P < 0.001]; and MPO, 5 ± 4 and 7 ± 1 at 28˚C and
32 ± 3 and 15 ± 2 at 37˚C in the CON and CAP groups, respectively, [$F_{(3,12)}$ = 85.62;
P < 0.001]). In mice exposed to an ambient temperature of 28˚C, there were no differences in
the counts of cFos-IR cells between the CON and CAP groups.

## Discussion

This study aimed to assess the effects of the desensitization of TRPV1 channels on heat-
escape/cold-seeking behavior. We observed that heat-escape/cold-seeking behavior was
completely blunted in the CAP group with an increase in $T_{abd}$, which reflected the core body
temperature. Exposure of the CAP group to 37˚C heat, during which heat-escape/cold-seeking
behavior was attenuated, resulted in a higher $T_{abd}$ and an attenuated increase in cFos expres-
sion in the POA as compared to that in the CON group.

Heat-escape/cold-seeking behavior is thought to contribute to thermoregulation [30]. To
assess this behavioral response, studies have evaluated the preference of animals for two floor
plates with different temperatures or thermal gradients [26, 27, 31]. The present system pre-
sented mice with a more complicated task to find their preferred board temperature. While
previous studies have reported resolution of the corner preference issue [32, 33], this study fur-
ther utilized Peltier boards, which can instantly change temperature of the area, thus, eliminat-
ing any location-related bias. During the control trial, all boards were set at a temperature of
32˚C, as mice exhibited a greater preference for this temperature than for other temperatures
used in our previous study [5]. $T_{abd}$ in both groups remained unchanged in the 32˚C control
trial (Fig 3A). Therefore, we conjectured that the 32˚C board did not provide a heat stimulus
that increased $T_{abd}$ in both groups. In addition, mice in both groups avoided resting on the
center board (Area 5). During the 36˚C, 38˚C, and 40˚C trials, mice in the CON group also
avoided Area 1 (one of the four corner boards), which deny the geographical influence for the
avoidance behavior. Moreover, mice in the CON group demonstrated clear escape behavior
when the board temperature was set at 36˚C, 38˚C, or 40˚C. These results suggest that the pres-
ent system can evaluate behavior in response to thermal stimuli, with decreasing geographical
influence on behavior.

In the CON group, the percentage of resting time on the 32˚C board (i.e., heat-escape/cold-
seeking behavior) increased linearly relative to the temperature of the other four boards (Fig
2). In contrast, the percentage of resting time at 32˚C in the CAP group was similar among the

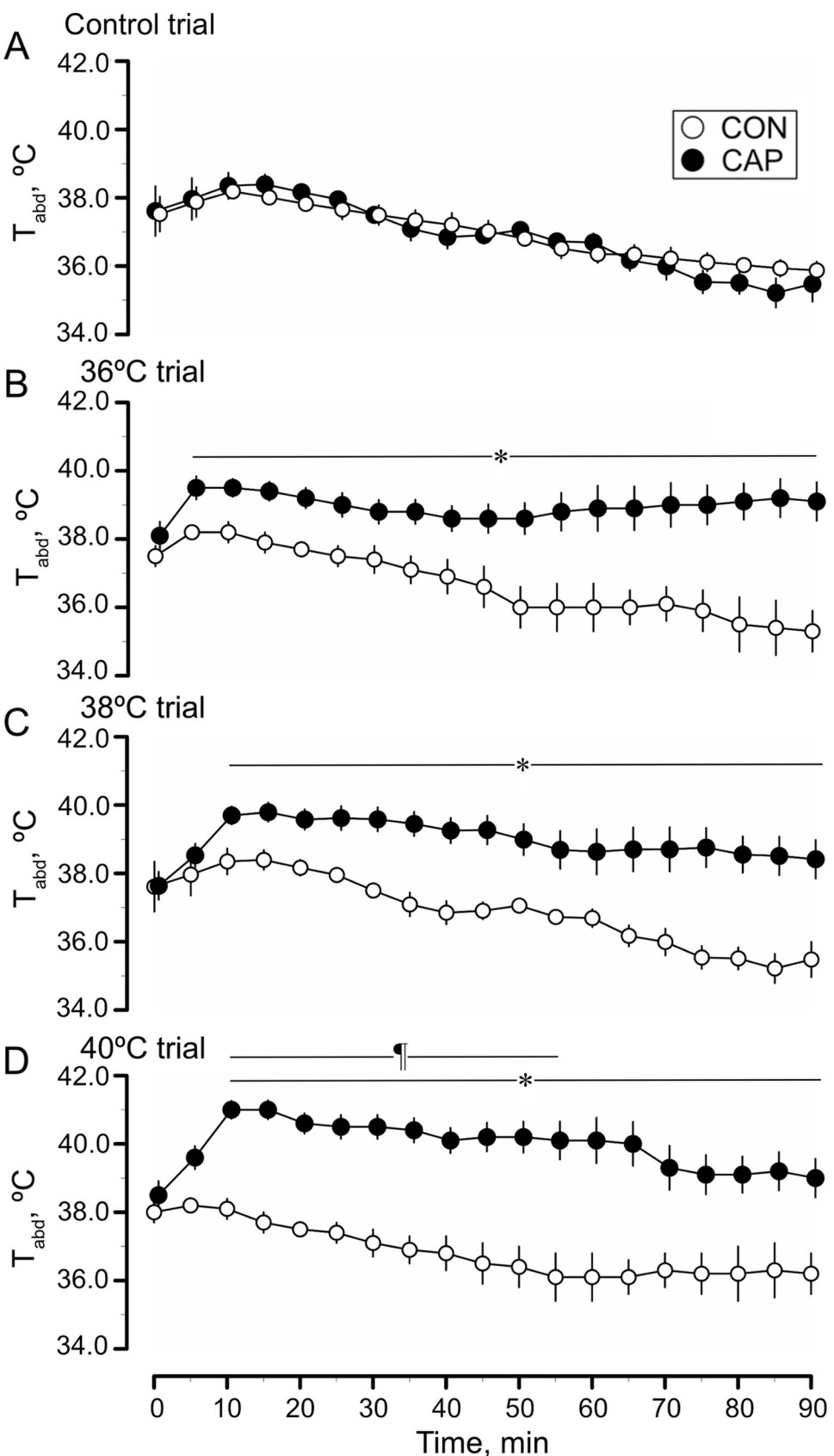

**Fig 3. Changes in intra-abdominal temperature (T$_{abd}$).** Changes in T$_{abd}$ during the control (n = 6), 36˚C (n = 4), 38˚C (n = 5 and 7, control and capsaicin groups, respectively), and 40˚C (n = 5 and 6, control and capsaicin groups, respectively) trials. Values are presented as means ± standard deviation (SD). *Significant difference between the control (CON) and capsaicin (CAP) groups, P < 0.05. ¶Significant difference compared to values during the 36˚C and 38˚C trials, P < 0.05.

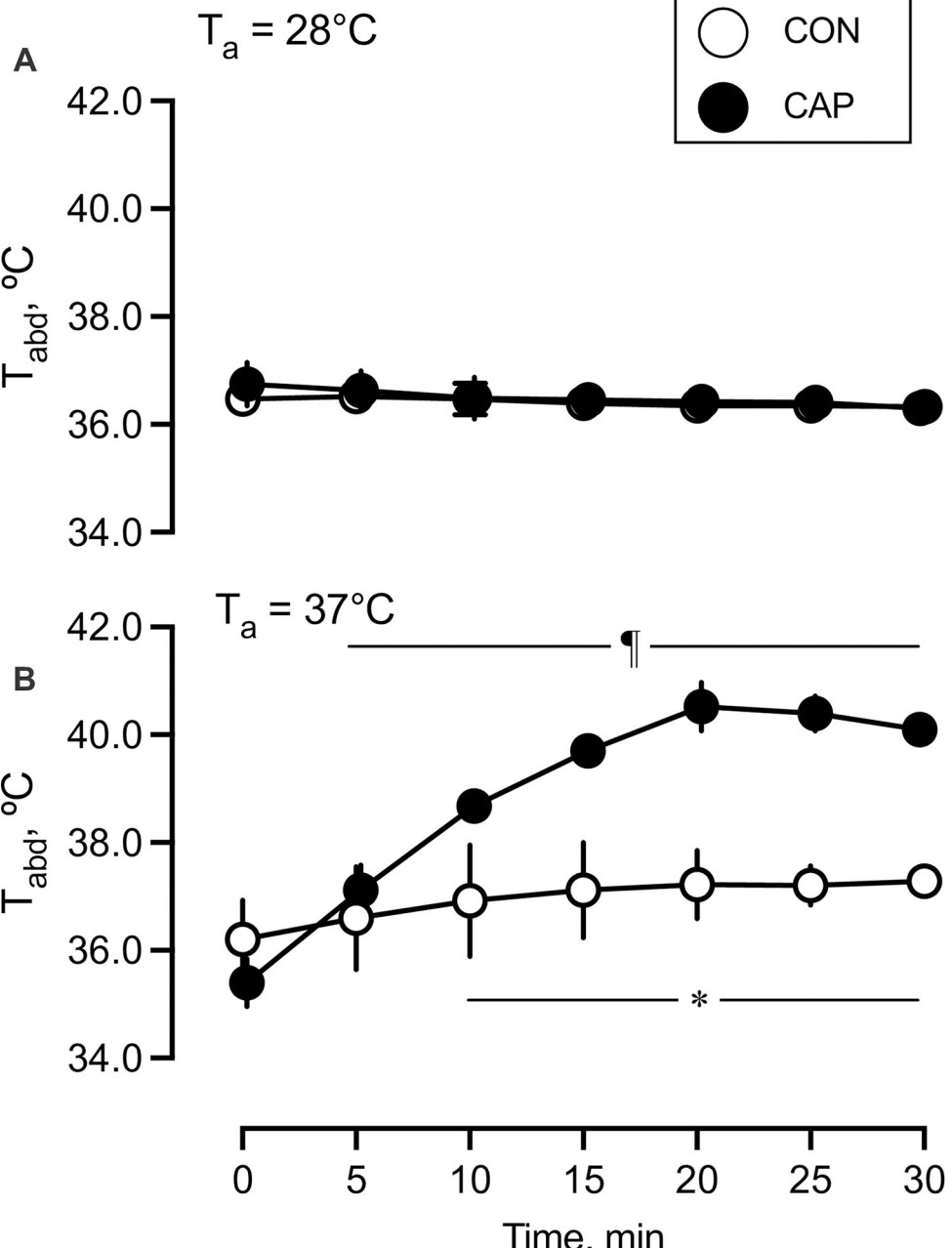

**Fig 4. Changes in intra-abdominal temperature (T$_{abd}$) during exposure to different ambient temperatures.** Changes in T$_{abd}$ during exposure to an ambient temperature of 28˚C (A) and 37˚C (B) in the control (CON) group (open circle; n = 4 and 5, 28˚C and 37˚C, respectively) and capsaicin (CAP) groups (closed circle; n = 4 and 5, 28˚C and 37˚C, respectively). Values are presented as means ± standard deviations (SD). ¶Significant difference compared to the value at 0 min. *Significant difference between the CON and CAP groups, P < 0.05.

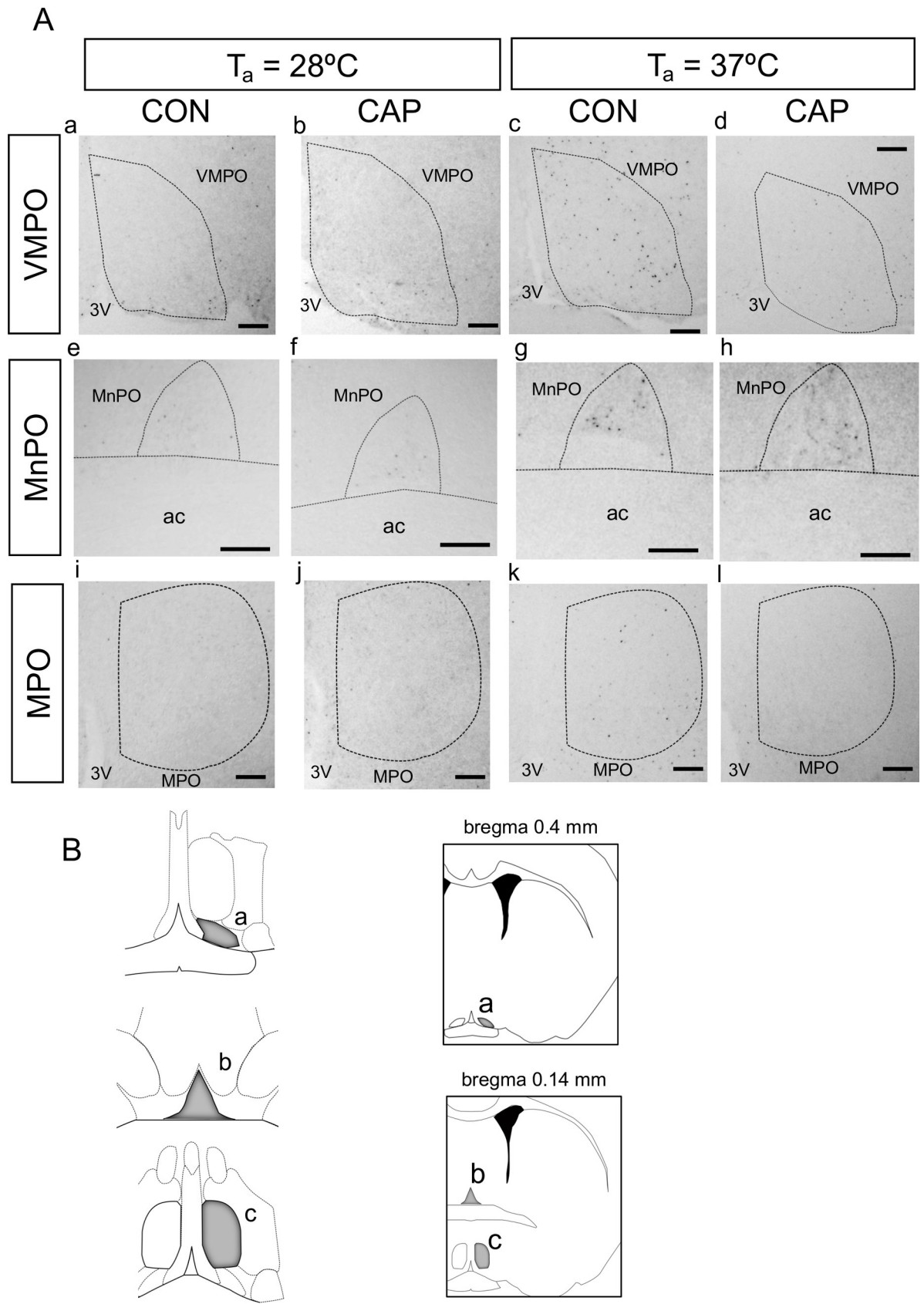

**Fig 5. Representative photoimages of cFos-immunoreactive (cFos-IR) cells.** Representative photoimages of cFos-IR cells in two sections (bregma 0.4 and 0.14 mm) in the preoptic area of the hypothalamus (POA) of the control (CON) group at $T_a$ of 28˚C (Aa, Ae, and Ai), capsaicin (CAP) group at $T_a$ of 28˚C (Ab, Af, and Aj), CON group at $T_a$ of 37˚C (Ac, Ag, and Ak), and CAP group at $T_a$ of 37˚C (Ad, Ah, and Al). The areas indicated in the dashed circles are the ventromedial POA (VMPO) (Fig 5Aa–5Ad), median POA (MnPO) (Fig 5Ae–5Ah), and medial POA (MPO) (Fig 5Ai–5Al). The areas are indicated as grey areas in the illustrations (Fig 5B; "a," "b," and "c" correspond to the VMPO, MnPO, and MPO, respectively), which were adapted from the Allen Mouse Brain Atlas [45]. Scale bar, 200 μm; 3V, third ventricle; ac, anterior commissure.

three trials. During the 36˚C, 38˚C, and 40˚C trials, the percentage of resting time at 32˚C was less than that in the CON group, with no difference among the three trials. Szolcsányi and Jancsó-Gábor examined thermal preferences in TRPV1 channel-desensitized rats injected with capsaicin using a two-chamber system with different ambient temperatures and reported that rats exhibited a greater preference for the 35˚C chamber than for the 30˚C chamber [34]. However, the control rats avoided the 35˚C chamber. Comparison of the 35˚C and 40˚C chambers revealed that desensitized rats avoided the 40˚C chamber but stayed in the 40˚C chamber for a longer duration compared to control rats. This phenomenon was explained as a result of the upward widening of the thermoneutral zone. The present study also demonstrated an altered ability of the CAP group to discern thermal stimuli. The reason for the different responses to high board temperatures in the present study remains unclear but may involve the effectiveness of capsaicin to achieve desensitization of TRPV1 channels. In this study, the thermoneutral zone of the CAP group could not be determined using the behavioral response results. In the control trial, $T_{abd}$ during the measurement period remained unchanged in both groups. In the 36˚C, 38˚C, and 40˚C trials, the CON group maintained the same $T_{abd}$ as that during the control trial; however, the CAP group became hyperthermic (Fig 3B–3D). In our previous study, we placed control mice on Peltier boards maintained at 39˚C and observed that $T_{abd}$ increased by 0.5–1.0˚C and plateaued within 90 min [5]. These results indicated that heat-escape/cold-seeking behavior plays a key role in thermoregulation (i.e., maintaining $T_{abd}$ during the 32˚C trial) under the present experimental conditions. Despite the higher $T_{abd}$ of the CAP group during the 36˚C, 38˚C, and 40˚C trials, $T_{abd}$ plateaued at approximately 39˚C at 60–90 min without any changes. Moreover, wet fur was not observed in any of the mice in either group. Therefore, a possible explanation for the plateauing of $T_{abd}$ is the partial activation of other thermoregulatory processes (i.e., autonomic thermoregulation), such as vasodilation of the tail and/or ear vessels. Capsaicin desensitization has been reported to affect autonomic and behavioral responses to heat [34]. We did not evaluate specific thermoregulatory responses other than heat-escape/cold-seeking behavior; however, the plateauing of $T_{abd}$ indicated that thermoregulatory processes were at least partly presented in the CAP group.

Tan et al. [8] identified warm-sensitive neurons in the POA of mice that were activated by exposure to an ambient temperature of 37˚C. When the neurons were locally stimulated by optogenetic techniques, mice became hypothermic due to increasing heat loss and decreasing heat production [8]. Stimulation of these neurons also activated cold-seeking behavior. In the present study, despite a greater increase in $T_{abd}$ during the 36˚C, 38˚C, and 40˚C trials in the CAP than in the CON group, heat-escape/cold-seeking behavior was not activated (Figs 2 and 3). A possible explanation is that heat-escape/cold-seeking behavior was activated by peripheral heat signals, which were abolished by the desensitization of TRPV1 channels. Another possibility is that the warm sensitivity of the POA was blunted by the direct action of capsaicin. Indeed, an attenuated response of the POA to body surface heat exposure in TRPV1 channel-desensitized animals was previously suggested [20]. Studies have reported that a dose of capsaicin (300 mg/kg) greater than that used in the present study (100 mg/kg) was necessary to inhibit the effect of local heating of the POA on autonomic vasodilation of the tail [18–20]. A

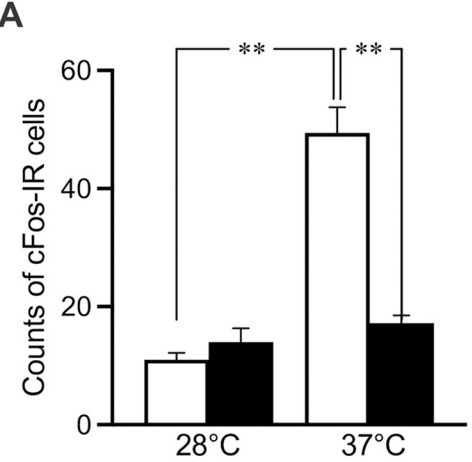

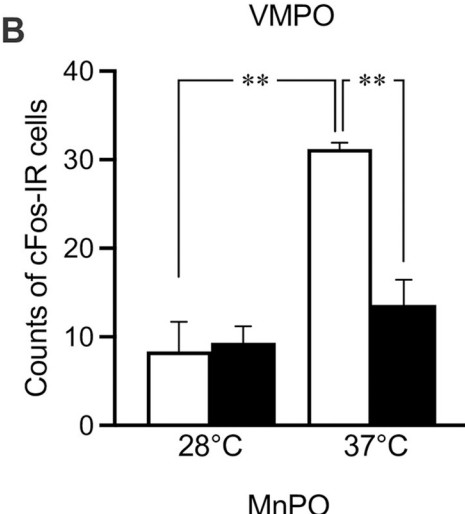

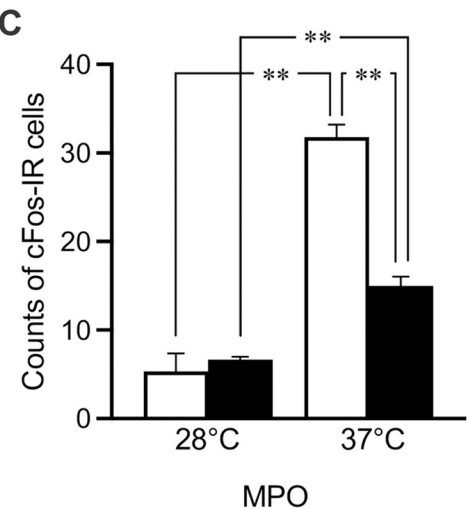

**Fig 6. cFos-immunoreactive (cFos-IR) cell counts.** Counts of cFos-IR cells in the ventromedial POA (VMPO) (A), median POA (MnPO) (B), and medial POA (MPO) (C) in the control (CON) (white bar; n = 3 and 5, at $T_a$ of 28°C and 37°C, respectively) and capsaicin (CAP) (black bar; n = 3 and 5, at $T_a$ of 28°C and 37°C, respectively) groups. Values are presented as means ± standard deviations (SD). **Significant difference between two corresponding trials, $P < 0.01$.

study by Molinas et al. [35] demonstrated abundant TRPV1 expression in the anterior, dorsomedial, and posterior regions of the hypothalamus. Thus, it is likely that the desensitization of TRPV1 channels was limited to the periphery in the present study. Additionally, the *trpv1* gene was reported to be necessary for thermal control of vasopressin release [36]. However, the role of TRPV1 in thermoregulatory responses in the hypothalamus has not been clearly delineated [37, 38]. Therefore, whether TRPV1 channel desensitization affects thermoregulation by modulating the POA remains debatable.

When mice were exposed to an ambient temperature of 37°C where heat-escape/cold-seeking behavior was not successfully available, $T_{abd}$ changed in a manner similar to that observed during Experiment 1 (Fig 4). The temperature of 37°C was selected to evaluate the influence of thermal input from the skin surface on thermoregulatory responses, minimizing that of core temperature. We reported that heat exposure at 39°C induced gradual increase in $T_{abd}$ within 60 min in mice [5]. Therefore, we assumed that the mice in the CON group could maintain $T_{abd}$ during the 30-min heat at 37°C (i.e., heat <39°C). In fact, $T_{abd}$ was unchanged in the CON group. The CAP group became hyperthermic, but $T_{abd}$ reached a plateau between 20 and 30 min. These results may suggest that thermoregulatory responses in the CAP group were maintained, but in a different manner from those in the CON group.

We evaluated cFos expression in subregions of the POA (i.e., MnPO, VMPO, and MPO), which relay thermal input from the skin. The number of cFos-IR cells in the POA of both groups increased after exposure to high ambient temperatures (Fig 5). However, the number of cFos-IR cells was higher in the CON group than in the CAP group. Cold signals from the body surface are relayed to the LPB in the pons and reach the median portion of the POA, including the MnPO [7, 39]. Furthermore, lesions of the LPB attenuate avoidance behavior with heat stimuli [7]. Moreover, the ventromedial part of the POA is activated by environmental heat, subsequently inducing heat-avoidance behavior [8]. In contrast, the effect of a TRPV1 agonist (i.e., capsaicin) on thermoregulation from peripheral stimulation remains controversial [40]. One study reported that a TRPV1 agonist only affected thermoregulation centrally, but not peripherally [41]. In this regard, desensitization induced by a TRPV1 agonist may have central effects. We did not use high-dose capsaicin, which can affect central desensitization. Additionally, peripheral capsaicin application has been reported to activate warm-activated neurons in the POA [8]. The POA contains neurons that respond to local warming of the hypothalamus [42] and are involved in various autonomic thermoregulatory responses [3]. Neurons expressing TRPM2 channels respond to local heating of the POA [43, 44]. Further, the POA contains neurons responsive to skin heating, which are observed even in TRPM2 channel knockout mice. The results suggest the presence of two distinct warm-sensitive neuronal populations in the POA that respond to thermal stimuli applied locally and to the skin. In the present study, we observed an increase in cFos-IR cells in the CON group despite a lack of increase in $T_{abd}$. Therefore, the number of cFos-IR cells in the POA may reflect neural activation induced by thermal input from the body surface and locally (in the POA). As $T_{abd}$ remained unchanged in the CON group, the number of cFos-IR cells may indicate the response to thermal input from the body surface. This speculation is supported by the greater number of cFos-IR cells in the MnPO in the CON group than in the CAP group (Fig 5). Moreover, the abundance of cFos-IR cells in the ventral part of the POA may reflect subsequent

neural activation of heat-escape/cold-seeking behavior. The sparse expression of cFos in the VMPO, MnPO, and MPO in the CAP group may reflect fewer thermal inputs from the body surface, which blunted efferent thermoregulatory pathway signaling related to thermoregulatory behavior. However, to verify this speculation, we need to evaluate the activation of other brain areas involved in the afferent neural pathway for thermoregulation, which should be also less activated in the CAP groups.

In the present study, we assessed thermoregulation in TRPV1 channel-desensitized mice. To our knowledge, this is the first study to use a newly developed system to demonstrate that heat-escape/cold-seeking behavior is a prominent thermoregulatory process in mice. This behavior was altered in TRPV1 channel-desensitized mice, possibly owing to the physiological blockade of TRPV1 channel-expressing neurons that transfer heat information from the body surface to the POA. However, other thermoregulatory responses were at least partially preserved. The behavioral response may be partly activated when thermal and heat-like (e.g., capsaicin) information from the body surface reaches the central nervous system. The desensitization of TRPV1 channels induced by capsaicin injection may be a valuable tool to understand the role of thermal information from the skin, visceral organs, and core during thermoregulation in environments with high ambient temperatures. However, the pharmacological mechanisms underscoring desensitization of TRPV1 channels remain unclear. In addition, there might be an influence of the capsaicin on the thermoregulatory central. Moreover, we did not assess the neurological response in the brain regions other than the POA that were involved in thermoregulatory efferents. Further investigations are warranted to fully elucidate the physiological mechanisms underpinning behavioral thermoregulation.

## Supporting information

**S1 Fig. Photoimages of cFos expression in the VMPO, MnPO, and MPO of the hypothalamus in the CON and CAP groups in Experiment 2.**
(PDF)

**S1 Table. Abdominal temperature in the CON and CAP groups in Experiment 2.**
(XLSX)

## Author Contributions

**Data curation:** Yuta Masuda, Riho Sakai.

**Formal analysis:** Yuta Masuda.

**Funding acquisition:** Kei Nagashima.

**Investigation:** Yuta Masuda, Riho Sakai, Issei Kato, Kei Nagashima.

**Supervision:** Kei Nagashima.

**Writing – original draft:** Yuta Masuda.

**Writing – review & editing:** Kei Nagashima.

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
