## [Decision Letter · Decision Letter 0]

4 Jul 2022

PONE-D-22-13073Thermoregulatory Heat-Escape/Cold-Seeking Behavior of Mice and the Influence of Capsaicin DesensitizationPLOS ONE

Dear Dr. Kei Nagashima,

Thank you for submitting your manuscript to PLOS ONE. After careful consideration, we feel that it has merit but does not fully meet PLOS ONE’s publication criteria as it currently stands. Therefore, we invite you to submit a revised version of the manuscript that addresses the points raised during the review process.

After reading the manuscript and considering the reports by the two reviewers, I believe it has merit, can be further evaluated, and possibly accepted for publication after several modifications to improve its quality. The first reviewer provided suggestions to improve the data interpretation and quality of the presentation, particularly the Results and Discussion sections. The second reviewer raised questions about methods, knowledge of relevant literature data, interpretation of the results, and the English language.

We look forward to receiving your revised manuscript.

Kind regards,

Samuel Penna Wanner, Ph.D.

Academic Editor

PLOS ONE

Journal Requirements:

“This work was supported by the W-ARCHS Research Project (A) for Establishing a Research Hub for Human Sciences.”

“•KN

•the W-ARCHS Research Project (A) for Establishing a Research Hub for Human Sciences.

•The funders had no role in study design, data collection and analysis, decision to publish, or preparation of the manuscript.”

Additional Editor Comments:

We have two outstanding reviews of the current manuscript that will help to improve its quality, particularly the interpretation of the results. Indeed, the second reviewer seems to provide more straightforward explanations for the current findings (considering the published literature) than the explanations included in the manuscript. In addition, we sometimes report findings that contradict the existing literature; although contradiction is part of advancing scientific knowledge, the existing literature must be acknowledged and discussed. This point should be addressed in the revised manuscript.

On another issue, the cover letter was written to Scientific Reports, not PLoS One. Please make the appropriate corrections in this letter and ensure that the manuscript is not currently under review in another journal.

I am sorry for taking too long to reach an editorial decision after receiving the comments from the reviewers. Unfortunately, I had COVID recently and stayed quarantined with my family, which restricted my working time. Regardless, I look forward to receiving a much-improved version of the current manuscript.

Reviewers' comments:

Reviewer's Responses to Questions

**Comments to the Author**

1. Is the manuscript technically sound, and do the data support the conclusions?

Reviewer #1: Yes

Reviewer #2: Partly

2. Has the statistical analysis been performed appropriately and rigorously? 

Reviewer #1: Yes

Reviewer #2: Yes

3. Have the authors made all data underlying the findings in their manuscript fully available?

Reviewer #1: Yes

Reviewer #2: Yes

4. Is the manuscript presented in an intelligible fashion and written in standard English?

Reviewer #1: Yes

Reviewer #2: No

5. Review Comments to the Author

Reviewer #1: This study was well conducted in methodological terms and the discussion was cautious for some interpretations of the results. The data were not entirely made available by the authors, however, the final results for the description of the results are show. A suggestion to insert some results that were not presented in the final manuscript was requested. The reading is easy to understand, just correcting some grammatical errors (listed in the review).

Reviewer #2: Masuda et al. developed a new device for the study of behavioral thermoregulation in rodents. In the newly developed system they show that the systemic capsaicin desensitization of mice impairs the heat escape/cold-seeking behavior and increases core temperature. A lower number of c-Fos-immunoreactive cells was also found in thermoregulatory nuclei in the preoptic area of the anterior hypothalamus in response to heat exposure after desensitization. The novel device and the topic of the paper is interesting, but questions related to methods, knowledge of relevant literature data, and interpretation of the results as well as their discussion should be addressed.

GENERAL COMMENTS

1) The authors miss to mention relevant findings from research groups which contributed with meaningful achievements to the fields of behavioral thermoregulation and the involved neural structures; of the role of TRPV1 in thermoregulation in rodents; and of the use thermogradient systems. This ignorance resulted in incorrect statements and conclusions in many places throughout the manuscript. Some major points are listed below:

i) the preoptic area is involved rather in autonomic than behavioral thermoregulation, while other brain areas such as the dorsal hypothalamic area. Was the c-Fos activity measured in that or other regions?

ii) while the exact role of TRPV1 in thermoregulation is still subject to research, some knowledge is already available about its function. For example, in rodents TRPV1 does not play a thermosensor role in thermoregulation, especially not in the periphery. Instead, TRPV1-expressing neurons in the MnPO mediate the effects of TRPV1 agonists and the same neuron population is presumably the one that is damaged by systemic vanilloid desensitization. Since the TRPV1-expressing MnPO neurons are within the warmth-defense thermoeffector pathway, their damage with systemic capsaicin desensitization can easily explain the bigger increase in core temperature in response to heat exposure, which was also observed by the authors in the present study. Because of the likely contribution of the damage of central TRPV1 channels to the observed changes, the discussion about the sole activation of cold-seeking behavior by peripheral heat signals (e.g., ln. 319-320; 336-338; and 354-356) should be rewritten. Also, when discussing the role of TRPV1 in thermoregulatory responses in the hypothalamus and the effect of capsaicin desensitization on POA neurons, the authors should get more insight into the literature of the field (collected, for example, in PMID: 19749171). In sum, the discussion about the role of peripheral signals should be much more limited since it is not supported at all with any evidence (as also stated by the authors in ln. 369-371). On the other hand, the potential damage of the hypothalamic (MnPO) neurons could be emphasized more as it is supported with scientific literature.

In relation with the above, it would be important to check if the lower c-Fos expression was specific to heat exposure or could it be due to a loss of neurons induced by capsaicin. Can the desensitized mice increase their c-Fos activity similarly as controls in the POA in response to stimuli other than heat? In either case, since the mice could not use heat escape as behavioral thermoregulation in the c-Fos experiments (Experiment 2), it is hard to understand how the observed changes indicate alterations in heat escape/cold-seeking.

It is a further related point that it is already known that autonomic warmth-defense responses (e.g., skin vasodilation) to heat are impaired in capsaicin-desensitized animals, therefore statements about their potential contribution to the plateau of Tb should be rephrased (ln. 306-309).

iii) when the authors compare their new system to previous devices (ln. 259-266), they mention that the corner preference was an obstacle in previous studies. However, they fail to mention a previously developed thermogradient system, which was repeatedly used for the study of behavioral thermoregulation (cold- and warmth-seeking) and could avoid the corner preference of the animals (see, for example, PMID: 16820025). The authors should also compare their new system to that one and familiarize with the results obtained in that system about brain nuclei of thermoregulatory behaviors.

SPECIFIC POINTS

1) Ln. 46-49: the authors should not refer to a study published in 2016 as recent.

2) Ln. 54-85: the last part of the sentence is unclear. Which two groups of mice experienced similar increases in body temperature?

3) The dose of capsaicin used for desensitization (50 + 50 mg/kg s.c.) is quite high. How was it tolerated by the mice? Did the authors use any intervention to help the recovery of the mice?

4) Why was it necessary to use different vehicle to dissolve capsaicin for desensitization and for the eye-wiping test?

5) Was always the same eye of the mice used in the eye-wiping experiments? Ln. 124: since the liquid were administered in a counterbalanced order, the application could not be randomized on the day of the second experiment.

6) Ln. 130: What does “Each system” mean? Were more than one systems built and used in the experiments?

7) Why was 32C set only in areas 2, 3, or 4, but not in area 1 during trials 1 to 3? Ln. 143-144: the authors state the order of the 4 trials was randomized, but later in Results this is contradicted as they state that the control trial was always the first. This should be explained.

8) It is probably enough to state that the results are expressed in the mean and SD format in line 181. So, it could be deleted each time the results are specified to improve the readability of the text (e.g., ln. 187; 194-195, etc.).

9) Ln. 215-217: was the resting time in area 5 also reduced compared to areas 1-4 in the desensitized mice?

10) Ln. 221-222: the sentence in its current form does not make sense, it should be corrected.

11) Ln. 234: the text says the difference started at 5 min, but in Fig. 4 it is shown from 10 min. It should be corrected.

12) Ln. 280-281: if we assume equal distribution among four areas, then why would be the ratio less than 25%? This does not make any sense. Accordingly, I find it misleading to discuss the results as if the increased warmth preference of the desensitized mice could be excluded (e.g., ln. 285-286), because that was not shown here. It was shown only that they spent less time than controls at 32C compared to 36, 38, and 40C. Similarly, I think it is also a false statement that their ability to discern thermal stimuli was completely abolished (ln. 294).

13) Ln. 344-346: “Another study” is mentioned, but both sentences refer to the same paper.

14) Ln. 349-351: The sentence looks incorrect. Why would unchanged temperature in the control group mean that the number of c-Fos cells indicate response to thermal stimuli from the body surface?

15) Ln. 364-365: The sentence in its current form is false. Thermoregulatory behavior can be activated by many other stimuli than “heat information”, for example, capsaicin, endotoxin, etc.

16) The quality of representative photos in Figure 5A should be checked. In printed version hardly any structure can be recognized. Bar scale should be also inserted.

17) The language of the manuscript should be substantially improved, e.g., ln. 38 “are are”; ln. 53 “..”; ln. 63 “TPPV1”; ln. 83: “intensityof”; ln. 221 “CAP and CAP groups”; ln. 242: “MnPO, MnPO”; ln. 253: capsicin; ln. 341 “han”, etc.

6. PLOS authors have the option to publish the peer review history of their article (what does this mean?). If published, this will include your full peer review and any attached files.

Reviewer #1: No

Reviewer #2: No

---

## [Author Response · Author response to Decision Letter 0]

30 Aug 2022

Responses to Reviewer #1

We thank the reviewer for the helpful comments and suggestions, which we have taken into consideration in our revised manuscript. The revised text is presented in blue in the manuscript.

Major comments

1. In the “Introduction”, lines 70-71, it was described that “It has been reported that systemic administration of larger doses of capsaicin results in the impairment of the hypothalamic function involved in thermoregulation”. Does the dose of 50 mg/kg (2 injections) induce only peripheral or central and peripheral desensitization?

>We conjecture that the desensitization was limited to the periphery, and the rationale for the conjecture in Ln 84-88, 410-413, and 415-416. 

2. In the Experiment 1, the animals could choose between the most thermally comfortable quadrant, where the smallest value is Ta = 32 °C (data obtained in previous results). Wouldn't an increase in Tabd be expected even in this temperature condition in both groups (CON and CAP)? In the trial at 40 °C, it was observed that, in the first 30 min, the CON group remained ~ 70% in 32 °C and then (60-90 min) this percentage rose to 90%. Would this percentage difference (with the mice remaining ~ 20% longer in the 40 °C at the first minutes) contribute to at least a transient rise in Tabd?

>We believe that the 32�C board did not provide a heat stimulus in both groups and added the rationale supporting this assumption in Ln 356 -358.

3. In the Experiment 1, CAP group showed an increase in Tabd in all 3 trials (36, 38 and 40 °C). The animals showed this increase in Tabd because: a- they lost the drive of peripheral temperature recognition and did not perform behavioral thermoregulation more effectively by staying longer at higher Ta (passive hyperthermia) or b- (making an addendum with experiment 2) was there an impaired c-Fos expression in APO, which in turn, prevented autonomic thermoregulation to dissipate heat in the face of increased Tabd? 

>We conjecture that option (a) is correct. The reason is that i) the desensitization of TRPV1 channels was limited to the periphery (Ln 84-88, 410-413, and 415-416), and we evaluated the cFos expression in the brain regions which may reflect the thermal input from the periphery and/or the central temperature, not thermoregulatory responses (Ln 429-430 and 437-450). In this regard, we also address the limitations of the present study in Ln 475-479.

4. In the Experiment 2, to induce passive heating, it was used a Ta = 37 °C; however, the CON group showed no increase in Tabd. Did the test fail to induce hyperthermia? Why was a new temperature value not selected?

>We confirmed that Ta above 37ºC induced unrecoverable hyperthermia, which precluded the discussion on autonomic thermoregulatory function given the increase in Tabd in the CON group. We have added an explanation for using a heat stress of 37°C (Ln 423-425), and the lack of the temperature signal result in the increase in Tabd in the CAP group (Ln 84-88, 410-413, and 415-416).

5. Overall, studies show that neuronal activation tends to be greater under conditions of increased core temperature. Interestingly, in this study, the CAP group, with higher values of Tabd, showed less neuronal recruitment and the authors attributed this response to the effect of capsaicin. Did the inhibition of neurons in APO by capsaicin prevent the activation of efferent pathways for heat dissipation, leading a more pronounced increase in Tabd?

>We added the rationale for the decreased activation of cFos (Ln 428-429 and 436-450). We did not evaluate the cFos expression which may reflect the efferent thermoregulatory pathway. The limitation of the present study not to be able to answer to your inquiry was also added (Ln 474-478). 

6. The authors attributed a certain maintenance of autonomic thermoregulation for heat dissipation since there is a plateau for Tabd increase in the CAP group. Was any other brain area (that contribute to autonomic thermoregulation) with a lower density of TRPV1 channels that could be analyzed to verify if heat stress promoted a different effect (i.e., increased c-Fos expression) and thus, to conclude that capsaicin selectively induced an inactivation of thermoreceptors in the APO?

>We focused on subregions in the POA that are associated with thermal input from the skin and did not consider other areas associated with an increase in central temperature. However, we cannot exclude the contribution of other brain areas, because the interaction of neural circuits involved in behavioral and autonomic thermoregulation remains unclear. We have added a relevant sentence accordingly (Ln 429-430, 437-451 and 475-479).

Minor comments

1. Capsaicin was used as a tool to desensitize the TRPV1 channels. Therefore, it may be more appropriate to replace the term “capsaicin” by “TRPV1” in the title and in other parts of the manuscript (i.e., Abstract, Aim…). For example: “Thermoregulatory Heat-Escape/Cold-Seeking Behavior of Mice and the Influence of TRPV1 channels”.

>We have replaced the term “capsaicin” by “TRPV1 channels” in the text as your suggestion suggested.

2. Abstract; Lines 21-22. “Mice were administered a subcutaneous injection of capsaicin (50 mg/kg; CAP group) for desensitization or vehicle (control [CON] group)”. Report what has been desensitized (TRPV1 channels).

>We have added this information to the revised text (Ln 22).

3. Abstract; Lines 24-27. The description of the methodology in the abstract needs to be further specified. It seems that only 1 experiment was done. However, 2 different experiments were carried out; one to evaluate the behavioral thermoregulation (exposition to boards system) and the other one, for autonomic thermoregulation (exposition to Ta = 37˚C).

>We have revised the sentences to clarify the methodology (Ln 23-34).

4. Abstract; Lines 27-31. To better understanding, results of experiment 1 (preference of different temperatures) and experiment 2 (changes in Tabd and c-Fos expression) should be describe separately. 

>The results for the two experiments have been described separately (Ln 23-34).

5. Introduction; Lines 41-43. The authors show a study where animals with hypothalamic injury had impaired automatic response to cold but maintained the behavioral response; however, they concluded that “this area of the brain may be responsible for these two different responses”. The conclusion contradicts the result! 

>We apologize for this error and have revised the sentence accordingly (Ln 49-52). 

6. Introduction; Line 61. “Studies have reported that local and/or systemic administration…”. Specify which location; in the brain? Which area?

>The administration was conducted via the subcutaneous tissue, aiming to desensitize the peripheral neurons expressing TRPV1 channels. Thus, we rewrote the sentence in Ln 73-75. 

7. Introduction; Lines 65-67. The authors cited 2 controversial studies: “It has been reported that capsaicin-desensitized rats prefer higher ambient temperature than normal rats (11)” and “Capsaicin-desensitized rats also showed avoidance behavior in 40°C heat (no reference!!!)”. They did not provide an explanation for these contradictory results. Is it necessary to cite 2 controversial studies in this moment? Does this information contribute to elaborate the rationale behind the study question?

>We wrote the rationale to cite the two studies and also added the missing reference in (Ln 77-82).

8. Introduction; Lines 74-77. I suggest merging sentences 1 and 2 in the same objective.

>The sentences have been merged as suggested (Ln 89-91).

9. Introduction; Lines 76-77. “We also compared the behavior of normal mice and capsaicin-desensitized mice”. The sentence became repetitive.

>The sentence has been removed.

10. Introduction; Lines 77-85. I suggest removing these sentences, once it is more applicated to “Methods” instead “Introduction”.

>We have removed these sentences as your suggestion.

11. Methods; Lines 115-126. In the context of desensitization of TRPV1 channels, what is meaning of "eye wipes"? It is important to mention in the manuscript.

We have revised this text to clarify the methodology and rationale for the behavior (134-146).

12. Methods; Line 154. Were the animals euthanized immediately after 30 minutes of passive heating? Was this time too short for evaluating the c-Fos expression?

>Animals were euthanized immediately after the 30-min heat. The period was determined based on the previous report (ref. 28). The part was changed (Ln 191-194).

13. Methods. It was not informed the coordinates used for sectioning the brain and the number of sections counted by area. 

This information has been added to the text (Ln 207-213).

14. Results; Lines 184-186. The first two sentences can be removed once they correspond to the Methods. Suggestion: “The desensitization of TRPV1 channels was evaluated using the test of eye wipes stimulated by local application of capsaicin.”

>The two sentences have been removed and replaced as suggested.

15. Results; Lines 190-192; 197-200. These sentences refer to the Methods. I suggest removing it from "Results".

>We have removed the sentences as suggested.

16. Figure 2. I suggest placing the legend at the top (next to graph A, instead of graph C). Inform N for each group.

>We have placed the legend at the top and defined N in the figure legend.

17. Figure 4. I suggest inserting the Tabd records at the 28 °C. These data would help to confirm the influence of TRPV1 channels only under high temperature conditions.

>We added the data of Tabd during the exposure at Ta of 28°C in Fig 4.

18. Figure 5. The scale of the photoimages was not shown.

>The scale was inserted in the photoimages in Fig 5 in addition the information of the length was added in the legend (Ln 330).

19. Figure 5; Graph C-b. The line that demonstrates the statistical difference is missing, as in the Graphs C-a and C-c.

>The line has been inserted accordingly.

20. Results; Lines 238-245. This information is best applied to the “figure legends”.

>The information has been moved to the legend of Fig 5 as suggested.

21. Figure 5. I suggest inserting the c-Fos expression at the 28 °C. Once again, these data would help to confirm the influence of TRPV1 channels only under high temperature conditions and it is important to evaluate the c-Fos expression in basal conditions and to compare with the stimuli (i.e., passive heating).

>Due to limited space in the figures, we have added the photoimages in Fig 5 and newly added Fig 6, which illustrates the count analysis and provided graphs of the quantitative analysis (Ln 322-337).

22. Inform the “N” in the legend of the figures and/or in the graphs.

>"N” has been defined in the figure legends at all instances. 

23. Discussion; Lines 251-254. The first sentence alone does not seem to be an objective of the study. The main focus is the desensitization of TRPV1 channels. Therefore, I suggest merging the first 2 sentences to define the objective of the study.

>The sentences have been merged and rewritten to clarify the main focus of this study (Ln 340-340).

24. Discussion; Lines 261-270. Re-evaluate the details showed in this section once they seem more applied to the “Methods” than to the “Discussion”. 

>This text has been revised and moved to the methods section (Ln 154-158).

25. Discussion, Lines 339-340. “The number of cFos-IR cells in the POA of both groups increased after exposure to high ambient temperatures”. This information was not cited in the “Results”. It was only reported that mice subjected to 28 °C showed less than 5; but it was not compared with the groups subjected to 37 °C.

>We have added relevant photoimages of cFos counts. Also, cFos quantitative analyses are presented in Fig 6 as a new graph.

26. Review typos. Some examples:

o Line 33: “(196 words)”.

o Line 38: “are are”.

o Line 27 and Line 37: standardize POA � preoptic area of the hypothalamus (POA) / preoptic area (POA) of the hypothalamus.

o Line 48: BDNF and PACAP were used only once in the manuscript. It is not necessary to use the acronym.

o Line 53: “of >40°C” � > 40 °C (check space); “nociception..” (2 points).

o Line 59: Yarmolinsky study was cited as reference of Caterina. Check references.

o Line 83: “intensityof”.

o Line 154: 1000 h.

o Lines 220-221. “Figure 3 shows Tabd during the control trial, the 36ºC trial (Fig 3A), the 38°C trial (Fig 3B), and the 40°C trial (Fig 3C) of the CAP and CAP groups”. Description is not in accordance with the figure. The correct form should be as follows: control trial (Fig. 3A), the 36ºC trial (Fig 3B), the 38°C trial (Fig 3C), and the 40°C trial (Fig 3D).

>Thank you for this helpful comment. The text has been corrected accordingly (Ln 275-276).

o Line 300: “hoewver".

o Line 341: “han”.

All the typos have been corrected.

 

Responses to Reviewer #2

We thank the reviewer for the helpful comments and suggestions, which we have taken into consideration in our revised manuscript. The revised text is presented in blue in the manuscript.

GENERAL COMMENTS

1) The authors miss to mention relevant findings from research groups which contributed with meaningful achievements to the fields of behavioral thermoregulation and the involved neural structures; of the role of TRPV1 in thermoregulation in rodents; and of the use thermogradient systems. This ignorance resulted in incorrect statements and conclusions in many places throughout the manuscript. 

Some major points are listed below:

i) the preoptic area is involved rather in autonomic than behavioral thermoregulation, while other brain areas such as the dorsal hypothalamic area. Was the c-Fos activity measured in that or other regions?

>>We focused on subregions in the POA that are associated with thermal input from the skin and did not consider other areas associated with an increase in central temperature. However, we cannot exclude the contribution of other brain areas, because the interaction of neural circuits involved in behavioral and autonomic thermoregulation remains unclear. We have added a relevant sentence accordingly (Ln 429-430, 437-451 and 475-479). We also added a description for our assumption that the brain regions of our interest reflects thermal inputs from the skin surface (Ln 45-47).

ii) while the exact role of TRPV1 in thermoregulation is still subject to research, some knowledge is already available about its function. For example, in rodents TRPV1 does not play a thermosensor role in thermoregulation, especially not in the periphery. Instead, TRPV1- expressing neurons in the MnPO mediate the effects of TRPV1 agonists and the same neuron population is presumably the one that is damaged by systemic vanilloid desensitization. Since the TRPV1-expressing MnPO neurons are within the warmth-defense thermoeffector pathway, their damage with systemic capsaicin desensitization can easily explain the bigger increase in core temperature in response to heat exposure, which was also observed by the authors in the present study. Because of the likely contribution of the damage of central TRPV1 channels to the observed changes, the discussion about the sole activation of cold-seeking behavior by peripheral heat signals (e.g., ln. 319-320; 336-338; and 354-356) should be rewritten. 

Also, when discussing the role of TRPV1 in thermoregulatory responses in the hypothalamus and the effect of capsaicin desensitization on POA neurons, the authors should get more insight into the literature of the field (collected, for example, in PMID: 19749171). In sum, the discussion about the role of peripheral signals should be much more limited since it is not supported at all with any evidence (as also stated by the authors in ln. 369-371). On the other hand, the potential damage of the hypothalamic (MnPO) neurons could be emphasized more as it is supported with scientific literature.

>We have revised our assertion in the text, given the limitation of explicitly stating that the phenomenon is purely due to peripheral desensitization, as you correctly pointed out. However, we believe that the peripheral effects of TRPV1 are controversial (Tan and Knight 2018). Indeed, after the study by Romanovsky et al. (2009), it has been demonstrated that peripheral TRPV1 agonist (capsaicin) administration activates warm-activated neurons in the POA (Tan et al. 2016). Furthermore, it has been reported that TRPV1 is very sparsely expressed centrally (in the brain) and to a lesser extent in the POA (Cavanaugh et al. 2011). Furthermore, the concentrations of capsaicin used in our study did not reach concentrations that would affect the central nervous system, as reported in previous studies (Pierau, Szolcsányi, and Sann 1986). We summarized these statements as a paragraph the discussion (Ln 430-464). 

In relation with the above, it would be important to check if the lower c-Fos expression was specific to heat exposure or could it be due to a loss of neurons induced by capsaicin. Can the desensitized mice increase their c-Fos activity similarly as controls in the POA in response to stimuli other than heat? In either case, since the mice could not use heat escape as behavioral thermoregulation in the c-Fos experiments (Experiment 2), it is hard to understand how the observed changes indicate alterations in heat escape/cold-seeking.

>We did not evaluate if other stimuli could increase the cFos expression. Although we did not have direct evidence, it is unlikely that capsaicin caused the loss of central neurons, since the dose used in this study did not reach a dose that would have affected the central nervous system, as reported in previous studies (Ln 84-88). In addition, we did not find any influence on food intake, body weight gain, and spontaneous movement, which reflect hypothalamic function in a part at least (Ln 131-132).

It is a further related point that it is already known that autonomic warmth-defense responses (e.g., skin vasodilation) to heat are impaired in capsaicin-desensitized animals, therefore statements about their potential contribution to the plateau of Tb should be rephrased (ln. 306- 309).

>We have rephrased this statement and discussed how capsaicin desensitization affects autonomic thermoregulation (Ln 392-398 and 437-443).

iii) when the authors compare their new system to previous devices (ln. 259-266), they mention that the corner preference was an obstacle in previous studies. However, they fail to mention a previously developed thermogradient system, which was repeatedly used for the study of behavioral thermoregulation (cold- and warmth-seeking) and could avoid the corner preference of the animals (see, for example, PMID: 16820025). The authors should also compare their new system to that one and familiarize with the results obtained in that system about brain nuclei of thermoregulatory behaviors. 

>We added the discussion in Ln 359-361.

SPECIFIC POINTS

1) Ln. 46-49: the authors should not refer to a study published in 2016 as recent. 

>The term “recent” has been removed.

2) Ln. 54-85: the last part of the sentence is unclear. Which two groups of mice experienced similar increases in body temperature?

>The part showing the hypotheses in the present study was revised (Ln 89-93).

3) The dose of capsaicin used for desensitization (50 + 50 mg/kg s.c.) is quite high. How was it tolerated by the mice? Did the authors use any intervention to help the recovery of the mice

>The intervention to minimize circulatory distress which is often observed after the injection was shown in Ln 122 - 125. In addition, the tolerance for the injection was added in Ln 130 – 132.

4) Why was it necessary to use different vehicle to dissolve capsaicin for desensitization and for the eye-wiping test? 

>The reason for the difference was because we followed the procedure for each previous report. We cited the respective study.

5) Was always the same eye of the mice used in the eye-wiping experiments? Ln. 124: since the liquid were administered in a counterbalanced order, the application could not be randomized on the day of the second experiment. 

>We have rewritten the sentence to clarify the procedure in Ln 144-145.

6) Ln. 130: What does “Each system” mean? Were more than one systems built and used in the experiments? 

>“Each” was replaced with “The” (Ln 151).

7) Why was 32C set only in areas 2, 3, or 4, but not in area 1 during trials 1 to 3? Ln. 143-144: the authors state the order of the 4 trials was randomized, but later in Results this is contradicted as they state that the control trial was always the first. This should be explained. 

>We have revised the sentences to clarify that the trials, including the control trial, were randomly ordered (Ln 159-171). The reason for setting area 1 at 36°, 38°C, or 40°C was to assess the geographic effects. However, we did not observe the effects. This has been added to the discussion, and the underlying rationale has been incorporated in the methods (Ln 168-167 and 358-361).

8) It is probably enough to state that the results are expressed in the mean and SD format in line 181. So, it could be deleted each time the results are specified to improve the readability of the text (e.g., ln. 187; 194-195, etc.). 

>The descriptions of SD have been removed.

9) Ln. 215-217: was the resting time in area 5 also reduced compared to areas 1-4 in the desensitized mice?

>We added a sentence indicating that the resting percentage at area 5 was also 0% and was less than that at other areas (Ln 270-272).

10) Ln. 221-222: the sentence in its current form does not make sense, it should be corrected.

>We corrected the sentence accordingly (Ln 275-276). 

11) Ln. 234: the text says the difference started at 5 min, but in Fig. 4 it is shown from 10 min. It should be corrected. 

>The part was corrected (Ln 299).

12) Ln. 280-281: if we assume equal distribution among four areas, then why would be the ratio less than 25%? This does not make any sense. Accordingly, I find it misleading to discuss the results as if the increased warmth preference of the desensitized mice could be excluded (e.g., ln. 285-286), because that was not shown here. It was shown only that they spent less time than controls at 32C compared to 36, 38, and 40C. Similarly, I think it is also a false statement that their ability to discern thermal stimuli was completely abolished (ln. 294). 

>We revised the part to clarify the interpretation of the data (Ln 365-378).

13) Ln. 344-346: “Another study” is mentioned, but both sentences refer to the same paper. 

>This was inadvertent error. We have replaced reference number 7 with number 8 (Tan CL, Cooke EK, Leib DE, Lin YC, Daly GE, Zimmerman CA, et al. Warm-sensitive neurons that control body temperature. Cell. 2016;167: 47–59.e15. doi: 10.1016/j.cell.2016.08.028.)

14) Ln. 349-351: The sentence looks incorrect. Why would unchanged temperature in the control group mean that the number of c-Fos cells indicate response to thermal stimuli from the body surface? 

>We conjecture that option (a) is correct. The reason is that i) the densitization of TRPV1 channels was limited to the periphery (Ln 84-88, 410-413, and 415-416), and we evaluated the cFos expression in the brain regions which may reflect the thermal input from the periphery and/or the central temperature, not thermoregulatory responses (Ln 429-430 and 437-451). In this regard, we also address the limitations of the present study in Ln 475-479.

15) Ln. 364-365: The sentence in its current form is false. Thermoregulatory behavior can be activated by many other stimuli than “heat information”, for example, capsaicin, endotoxin, etc.

>The sentence has been reworded accordingly (Ln 470-472).

16) The quality of representative photos in Figure 5A should be checked. In printed version hardly any structure can be recognized. Bar scale should be also inserted. 

>The images were replaced, and scale bars were inserted.

17) The language of the manuscript should be substantially improved, e.g., ln. 38 “are are”; ln. 53 “..”; ln. 63 “TPPV1”; ln. 83: “intensityof”; ln. 221 “CAP and CAP groups”; ln. 242: “MnPO, MnPO”; ln. 253: capsicin; ln. 341 “han”, etc. 

>The text has been corrected accordingly, thank you.

---

## [Decision Letter · Decision Letter 1]

29 Sep 2022

PONE-D-22-13073R1Thermoregulatory heat-escape/cold-seeking behavior in mice and the influence of TRPV1 channelsPLOS ONE

Dear Dr. Kei Nagashima,

Thank you for submitting your manuscript to PLOS ONE. After careful consideration, we feel that it has merit but does not fully meet PLOS ONE’s publication criteria as it currently stands. Therefore, we invite you to submit a revised version of the manuscript that addresses the points raised during the review process. The authors satisfactorily addressed most of the comments made by the two reviewers. Thank you. As a result, the revised manuscript was much improved compared to the original one. However, both reviewers have indicated that minor corrections are still required before the manuscript is ready for publication. In addition, please see my comments at the end of this letter and make the required changes.

We look forward to receiving your revised manuscript.

Kind regards,

Samuel Penna Wanner, Ph.D.

Academic Editor

PLOS ONE

Journal Requirements:

Additional Editor Comments:

1. Abstract, Line 26. Please place the second comma after the word “board” in the following sentence: “… four boards, including the center, board were set …”.

2. Results, Line 247. Please replace “tha” with “that”.

3. Results, Lines 314 to 318. Three experimental groups were mentioned in the sentence, but there are data (averages and standard deviations) of four groups inside the parentheses. The authors may want to edit this sentence to improve clarity. This edit may be critical because the figure will not necessarily be close to/on the same page as the graph in the edited manuscript.

4. Please check if any additional clarification should be made to address the two major points that are still “bothering” the first reviewer. Concerning his/her first point, the abdominal temperature of the control group increased by approximately 1oC at 37oC (Figure 4B). The lack of statistical significance may result from the low number of animals used in this experiment (n = 5). The authors may want to comment on this point.

5. The first reviewer also listed five suggestions (i.e., minor points). Please adjust the manuscript accordingly to these suggestions. Note that, in the fourth comment, it should be figure 3 instead of figure 5.

6. The second reviewer still asks for clarification regarding the randomization of treatments in the eye wiping test. Please clarify this issue in the revised manuscript.

7. The second reviewer also mentioned that he/she could not open the data uploaded in the supplementary material. I could not open the file named “＊Photoimages_R1.pdf” either. Please double check if everything is correct with supplementary material.

8. If the authors address these comments satisfactorily, the manuscript can be accepted for publication without another (i.e., third) round of external reviews.

Reviewers' comments:

Reviewer's Responses to Questions

**Comments to the Author**

1. If the authors have adequately addressed your comments raised in a previous round of review and you feel that this manuscript is now acceptable for publication, you may indicate that here to bypass the “Comments to the Author” section, enter your conflict of interest statement in the “Confidential to Editor” section, and submit your "Accept" recommendation.

Reviewer #1: All comments have been addressed

Reviewer #2: (No Response)

2. Is the manuscript technically sound, and do the data support the conclusions?

Reviewer #1: Yes

Reviewer #2: Yes

3. Has the statistical analysis been performed appropriately and rigorously? 

Reviewer #1: Yes

Reviewer #2: Yes

4. Have the authors made all data underlying the findings in their manuscript fully available?

Reviewer #1: Yes

Reviewer #2: Yes

5. Is the manuscript presented in an intelligible fashion and written in standard English?

Reviewer #1: Yes

Reviewer #2: Yes

6. Review Comments to the Author

Reviewer #1: All the comments were described in the archive.

xxxxxxxxxxxxxxxxxxxxxxxxxxxxxxxxxxxxxxxxxxxxxxxxxxx

Reviewer #2: The authors responded to my comments in a satisfactory manner. One minor issue that I still do not understand is related to randomization of treatments in the eye wiping test: if two substances are used (capsaicin and vehicle) and one of them is applied first and the other one 2 days later, then it does not look possible to randomize the order of application on both days (ln. 144-145). The treatment applied first will automatically determine the second treatment, hence no randomization can be performed at the second administration.

7. PLOS authors have the option to publish the peer review history of their article (what does this mean?). If published, this will include your full peer review and any attached files.

Reviewer #1: No

Reviewer #2: No

---

## [Author Response · Author response to Decision Letter 1]

7 Oct 2022

Additional Editor Comments:

1. Abstract, Line 26. Please place the second comma after the word “board” in the following sentence: “… four boards,including the center, board were set …”.

As per the editor’s insightful suggestion, we have placed the second comma after the word “boards.” 

(Line 26)

2. Results, Line 247. Please replace “tha” with “that”.

We would like to apologize to the editor for this mistake. We have corrected this mistake in the revised manuscript (Line 246).

3. Results, Lines 314 to 318. Three experimental groups were mentioned in the sentence, but there are data (averages and standard deviations) of four groups inside the parentheses. The authors may want to edit this sentence to improve clarity. This edit may be critical because the figure will not necessarily be close to/on the same page as the graph in the edited manuscript.

We would like to thank the editor for this valuable suggestion. Please note that we have revised this part to present the data of four groups. The revised part is as follows:

“In the three areas, the counts were greater in the CON group at 37°C than those in the CON group at 28°C and CAP group at 37°C (VMPO; 11 ± 2 and 14 ± 4 at 28˚C and 49 ± 10 and 17 ± 3 at 37˚C in the CON and CAP groups, respectively, [F(3,12) = 36.77; P < 0.001]; MnPO, 8 ± 6 and 9 ± 3 at 28˚C and 31 ± 2 and 14 ± 6 at 37˚C in the CON and CAP groups, respectively, [F(3,12) = 22.27; P < 0.001]; and MPO, 5 ± 4 and 7 ± 1 at 28˚C and 32 ± 3 and 15 ± 2 at 37˚C in the CON and CAP groups, respectively, [F(3,12) = 85.62; P < 0.001]). In mice exposed to an ambient temperature of 28°C, there were no differences in the counts of cFos-IR cells between the CON and CAP groups.” (Lines 315–322)

4. Please check if any additional clarification should be made to address the two major points that are still “bothering” the first reviewer. Concerning his/her first point, the abdominal temperature of the control group increased by approximately 1°C at 37°C (Figure 4B). The lack of statistical significance may result from the low number of animals used in this experiment (n = 5). The authors may want to comment on this point.

5. The first reviewer also listed five suggestions (i.e., minor points). Please adjust the manuscript accordingly to these suggestions. Note that, in the fourth comment, it should be figure 3 instead of figure 5.

Please note that we have responded to the reviewer in our response to his/her Comment #1.

6. The second reviewer still asks for clarification regarding the randomization of treatments in the eye wiping test. Please clarify this issue in the revised manuscript.

As per the editor’s insightful suggestion, we have revised this part as follows:

“Liquid (i.e., capsaicin solution or the vehicle) and eye laterality (i.e., right or left) on Day 1 were randomly selected.” (Lines 143–144)

7. The second reviewer also mentioned that he/she could not open the data uploaded in the supplementary material. I could not open the file named “＊Photoimages_R1.pdf” either. Please double check if everything is correct with supplementary material.

Please note that we have re-uploaded the data and verified that we could open the file. 

Responses to Reviewer #1

Major point:

The most important questions that still lack a plausible explanation are the fact that: 

(1) the CON group did not show an increase in Tabd at relatively high Ta, i.e., 32°C (in the Experiment 1) and, mainly, 37 °C (in the Experiment 2)

To the first point, the authors informed that the Ta = 37°C was chosen to avoid unrecoverable hyperthermia caused by higher ambient temperature than 37°C and to evaluate thermoregulatory responses including autonomic system responses. I understand that this temperature value did not induce an increase in Tabd in the CON group (unlike what was explained) and, even so, an autonomic response is expected in these conditions. So I think the “Experiment 2” failed to induce passive hyperthermia as proposed.

The authors would like to thank the reviewer for his/her constructive critique to improve the manuscript. We have made every effort to address the issues raised and to respond to all comments. The revisions are indicated in blue font in the revised manuscript. Please, find next a detailed, point-by-point response to the reviewer's comments. We hope that our revisions will meet the reviewer’s expectations.

We agree with the reviewer that Tabd was maintained in the CON group via autonomic thermoregulatory response. To clarify this point and show the rationale for the selection of heat exposure at 37˚C, we have added the following part to the Discussion section of the revised manuscript:

“When mice were exposed to an ambient temperature of 37°C where heat-escape/cold-seeking behavior was not successfully available, Tabd changed in a manner similar to that observed during Experiment 1 (Fig 4). The temperature of 37°C was selected to evaluate the influence of thermal input from the skin surface on thermoregulatory responses, minimizing that of core temperature. We reported that heat exposure at 39°C induced gradual increase in Tabd within 60 min in mice [5]. Therefore, we assumed that the mice in the CON group could maintain Tabd during the 30-min heat at 37°C (i.e., heat <39°C). In fact, Tabd was unchanged in the CON group. The CAP group became hyperthermic, but Tabd reached a plateau between 20 and 30 min. These results may suggest that thermoregulatory responses in the CAP group were maintained, but in a different manner from those in the CON group.” (Lines 427–437) 

(2) the reduction of neuronal activation in the APO in the CAP group even with the hyperthermic response.

To the second point, the authors attribute the lower neuronal activation to the possible reduction of afferent inputs that were reduced by peripheral desensitization of TRPV1 channels, considering that the capsaicin treatment would not change the central TRPV1 channels. It would be important to evaluate other hyperthermia-responsive nuclei to verify a possible increase in activation and confirm the hypothesis.

As per the reviewer’s suggestion, we have discussed this issue as a limitation as follows:

“However, to verify this speculation, we need to evaluate the activation of other brain areas involved in the afferent neural pathway for thermoregulation, which should be also activated in the CAP groups.” (Lines 469–472)

Minor points:

1. Introduction (line 47): After citing reference 4, start the text with a new paragraph and introduce a link to start contextualizing behavioral thermoregulation. "On the other hand, in the behavioral thermoregulation, the heat-escape cold-seeking behavior is a key…"

Please note that we have revised this part as per the reviewer’s suggestion. The revised part is as follows:

“In contrast, in behavioral thermoregulation, heat-escape/cold-seeking behavior is a key thermoregulatory response to heat [5] and is rapidly initiated after heat exposure [2].” (Lines 48–50)

2. Introduction (lines 51 and 54): I suggest that the conclusions of cited studies follow in the same sentence of the description of the results.

Please note that we have revised this part as per the reviewer’s suggestion. The revised part is as follows:

“Satinoff and Rutstein reported that the autonomic cold response was impaired after the lesion of the anterior hypothalamus in rats but the behavioral response was retained [6], which denies an involvement of the anterior hypothalamus in the observed behavioral response. Additionally, the lesion of the LPB, receiving thermal inputs from the body surface, abolished heat-avoidance behavior in rats [7], indicating the necessity of only peripheral thermal signals to activate behavioral responses.” (Lines 50–55)

3. Introduction (lines 77 and 79): It seemed confusing to inform the values of Ta in parentheses as if they were referring to the rats.

Please note that we have revised this part as per the reviewer’s suggestion. The revised part is as follows:

“Although the control rats prefer ambient temperature of 30˚C, rats with capsaicin-induced desensitization of TRPV1 channels prefer higher ambient temperature (35°C) [13].” (Lines 77–79)

4. Legend of Figure 53: The symbol † does not appear in the graph. Remove it of legend.

Please note that we have removed this symbol from the legend, as per the reviewer’s suggestion.

5. Legend of Figure 5: Change “Ab, d and f” � “Ab, f and j.”

Please note that we have changed the legend as per the reviewer’s comment (Line 328).

---

## [Editor Report · Decision Letter 2]

13 Oct 2022

Thermoregulatory heat-escape/cold-seeking behavior in mice and the influence of TRPV1 channels

PONE-D-22-13073R2

Dear Dr. Kei Nagashima,

We’re pleased to inform you that your manuscript has been judged scientifically suitable for publication and will be formally accepted for publication once it meets all outstanding technical requirements.

Kind regards,

Samuel Penna Wanner, Ph.D.

Academic Editor

PLOS ONE

Additional Editor Comments (optional):

After reading the revised manuscript, I believe the authors have adequately addressed all the minor points the two reviewers and I (i.e., the Academic editor) have raised. Thank you! The manuscript deserves to be published in PLOS One in its current form. Congratulations.

If the authors get a chance, please check if the sentence in line 471 lacks the word “less” before the word “activated”.
---

## [Editor Report · Acceptance letter]

7 Nov 2022

PONE-D-22-13073R2 

Thermoregulatory heat-escape/cold-seeking behavior in mice and the influence of TRPV1 channels 

Dear Dr. Nagashima:

I'm pleased to inform you that your manuscript has been deemed suitable for publication in PLOS ONE. Congratulations! Your manuscript is now with our production department. 

Kind regards, 

on behalf of

Dr. Samuel Penna Wanner 

Academic Editor

PLOS ONE